# Age and Sex Determine Electrocardiogram Parameters in the *Octodon degus*

**DOI:** 10.3390/biology12050747

**Published:** 2023-05-19

**Authors:** Lorena Cuenca-Bermejo, María Josefa Fernández-Del Palacio, Valeria de Cassia Gonçalves, Víctor Bautista-Hernández, Consuelo Sánchez-Rodrigo, Emiliano Fernández-Villalba, Karolina Kublickiene, Valeria Raparelli, Alexandra Kautzky-Willer, Colleen M. Norris, Louise Pilote, María Trinidad Herrero

**Affiliations:** 1Clinical & Experimental Neuroscience (NiCE), Biomedical Research Institute of Murcia (IMIB-Pascual Parrilla), University of Murcia, 30120 Murcia, Spain; lorena.cuenca@um.es (L.C.-B.); vaal.cassia@gmail.com (V.d.C.G.); consuelo.sanchezr@um.es (C.S.-R.); emiliano.fernandez@um.es (E.F.-V.); 2Institute for Aging Research (IUIE), Campus Mare Nostrum, European University for Wellbeing (EUniWell), 30100 Murcia, Spain; 3Department of Animal Medicine and Surgery, Veterinary Teaching Hospital, Campus de Espinardo, University of Murcia, 30100 Murcia, Spain; 4Disciplina de Neurociência, Departamento de Neurologia e Neurocirurgia, Universidade Federal de São Paulo (UNIFESP), São Paulo 04039-032, Brazil; 5Congenital Heart Surgery, Baylor College of Medicine, Houston, TX 77030, USA; victor.bautista-hernandez@christushealth.org; 6Division of Renal Medicine, Department of Clinical Science, Intervention and Technology, Karolinska Institutet, 17177 Stockholm, Sweden; karolina.kublickiene@ki.se; 7Department of Translational Medicine, University of Ferrara, 44124 Ferrara, Italy; valeria.raparelli@unife.it; 8University Center for Studies on Gender Medicine, University of Ferrara, 44121 Ferrara, Italy; 9Division of Endocrinology and Metabolism, Department of Medicine III, Medical University of Vienna, 1090 Vienna, Austria; alexandra.kautzky-willer@meduniwien.ac.at; 10Faculty of Nursing, University of Alberta, Edmonton, AB T6G 2R3, Canada; colleen.norris@ualberta.ca; 11Cardiovascular and Stroke Strategic Clinical Network, Alberta Health Services, Edmonton, AB T5J 3E4, Canada; 12Division of Clinical Epidemiology, Research Institute of McGill University Health Centre, McGill University, Montreal, QC H4A 3J1, Canada; louise.pilote@mcgill.ca

**Keywords:** electrocardiogram, *Octodon degus*, aging, sex, heart, arrhythmia

## Abstract

**Simple Summary:**

Cardiovascular diseases represent the leading cause of mortality and morbidity worldwide, and age is an important contributing factor. Preclinical models are crucial for exploring and understanding the physiology of the heart and age-related changes, as well as cardiac pathological aspects. In line with this, the *Octodon degus* (*O. degus*) is a diurnal rodent that has been claimed as an interesting model for aging research. However, several aspects of this experimental model are not fully characterized yet. In this work, we evaluated the cardiac electric activity by electrocardiogram recording in the aging process, considering differences between males and females. We provide the normal ranges for the heart rate, duration and voltage of the ECG waves and intervals, as well as electrical axis deviation. Our results demonstrate that both age and sex influence the cardiac function and the incidence of arrhythmias. Therefore, this research indicates that this rodent model could be useful for cardiovascular research, including impacts of aging and biological sex.

**Abstract:**

Cardiovascular diseases represent the leading cause of mortality and morbidity worldwide, and age is an important risk factor. Preclinical models provide supportive evidence toward age-related cardiac changes, as well as allow for the study of pathological aspects of the disease. In the present work, we evaluated the electrocardiogram (ECG) recording in the *O. degus* during the aging process in both females and males. Taking into account the age and sex, our study provides the normal ranges for the heart rate, duration and voltage of the ECG waves and intervals, as well as electrical axis deviation. We found that the QRS complex duration and QTc significantly increased with age, whereas the heart rate significantly decreased. On the other hand, the P wave, PR and QTc segments durations, S wave voltage and electrical axis were found to be significantly different between males and females. The heart rhythm was also altered in aged animals, resulting in an increased incidence of arrhythmias, especially in males. Based on these results, we suggest that this rodent model could be useful for cardiovascular research, including impacts of aging and biological sex.

## 1. Introduction

In the last few decades, the worldwide aged population (>65 years old) and the life expectancy have unprecedently increased [1]. Unfortunately, this increase in the life expectancy of the population is associated with the higher impact and prevalence of age-associated disorders, cardiovascular diseases being the leading cause of death in high-income countries [2]. Importantly, cardiac changes can be prodromal to other diseases associated with aging, thereby influencing their progression [3,4,5,6]. Current data point out that sex differences play a key role in cardiovascular diseases along with aging, although how their interaction may influence different clinical outcomes remains unclear [7]. Thus, the study of age-related changes in cardiac function is required to better understand how physiological heart aging could, together with other factors, predispose patients to pathological conditions. This approach would contribute to the elaboration of better guidelines to manage the patient’s diagnosis, prognosis, therapeutical design and quality of life.

A surface electrocardiogram (ECG) is one of the most commonly used methods for evaluating heart physiology, both in humans and animals [8]. Indeed, ECG is the most important clinical test for the interpretation of the cardiac rhythm, detection of myocardial ischemia and infarction, preexcitation, long QT syndromes and difficulties in the electrical transmission between the different parts of the heart [9,10]. In animal models, several studies have reported that ECG parameters in healthy and diseased animals are influenced by different experimental conditions, including aging, myocardial infarction or systemic arterial hypertension [11,12,13,14].

Research in human subjects is associated with several obstacles and, therefore, the use of animal models that mimic the human aging process and its related disorders is crucial. In line with this, the diurnal rodent *Octodon degus* (*O. degus*) has been proposed as a relevant tool in biomedical research because some of its features are similar to humans, including its complex social behavior, circadian rhythm and genetic homologies. In addition, aged degus show traits shared with human aging, such as variations in hematological parameters, brain pathology (e.g., cognitive decline, Alzheimer’s disease-like histopathological markers, neuroinflammation), degeneration of the visual system, type II diabetes, development of spontaneous tumors, disruption of lipid metabolism, atherosclerosis and renal problems, among others [15]. However, studies evaluating the cardiovascular system in the *O. degus* are scarce [16] and no current studies have reported data on the electrical activity of the heart of this rodent. Considering that this model is susceptible to different age-related diseases, we hypothesize that the ECG parameters in *O. degus* are influenced by the age and sex and thus could be used as a natural model for age-associated cardiac alterations. Hence, the aims of this study were to determine the normal surface ECG parameters in the *O. degus* and to evaluate the effect of age and sex on them.

## 2. Materials and Methods

### 2.1. Animals and Experimental Groups

The present study was performed on a total of 98 *O. degus* that were divided into experimental groups according to sex (males and females) and age: (i) 6 months old (juvenile), (ii) 1 year old (young), (iii) 4–5 years old (old) and (iv) 6–7 years old (senile) (Table 1) [17]. Animals belonged to the colony of our research group. They were bred and maintained in the animal house of the University of Alicante and in the Institute for Biomedical Research of Murcia (CEIB-IMIB-University of Murcia) under identical conditions. Animals were individually identified by microchips and housed in groups of 2 to 4 animals per cage in a room with controlled environmental conditions: 12:12 h light: dark cycles, 22 ± 1 °C and 60% humidity. The cages, which were cleaned once per week, were made of Plexiglas, their floor was covered with wood shavings and environmental enrichment was placed on them. Animals had water and food ad libitum during the entire experiment (Harlan Tekland Global Diet^®^, Harlan Laboratories, USA).

This study was part of a project that considers the study of different postmortem parameters. For this purpose, animals were sacrificed with a CO_2_ flux and, during the sample collection, hearts were weighted.

### 2.2. Ethics Statement

All experimental procedures involving animals were performed strictly following the regulations standards indicated in experimental guidelines and procedures complying with the 1964 Declaration of Helsinki (and its later amendments), the European Community Council Directive (2010/63/UE), the Spanish law (RD 53/2013) and the Ethics Committee of the University of Murcia (project number: A13170102/CEEA-OH AMP/MOD 103/2014 + 2018). Additionally, the “3 Rs principle” was applied throughout the study and important efforts were made to minimize the procedure’s impact on animals. All the animals used in this study underwent handling by the experimenters (at least 5 min per day) to accustom them to manipulation and to reduce the stressful conditions. The health status of animals was continuously examined by an expert veterinarian. Animals included in this study did not have any evident alteration at physical examination.

### 2.3. Electrocardiogram Recordings

The ECG recordings were carried out in the facilities of the Veterinary Teaching Hospital of the University of Murcia using a commercially available 6-channel ECG machine (Siemens Megacart R, Electromedical Systems Division ECS, Solna, Sweden).

Animals were placed in an isolated room where the ECG recording would be performed at least 8 h before the test started, and they were kept undisturbed in order to reduce stress and promote acclimatization. All the ECGs were always performed at fixed times (between 4:00–8:00 p.m.) to reduce the influence of circadian variables during the analysis. Animals were anesthetized with an intramuscular combination of 20.0 mg/kg of ketamine (Anesketin^®^, Dechra Veterinary Products SLU, Barcelona, Spain) and 0.20 mg/kg of medetomidine (Domtor^®^, Ecuphar^®^, Barcelona, Spain).

All *O. degus* were placed in right lateral recumbency, with the limbs held as nearly perpendicular to the body as possible. Surface electrodes made of flattened alligator clips were attached to the skin midway on the caudal aspect of the elbows and over the patellae on the rear limbs (Figure 1A). Surgical alcohol was applied to maintain proper electrical contact. Standard bipolar and augmented unipolar 6-lead ECGs (leads I, II, III, aVR, aVL and aVF) were recorded for all animals. According to heart rate (HR) and amplitude of the waves, the ECG recordings were calibrated at 50–100 mm/s and 10–20 mm/mV. For each animal, ECG tracing was obtained with a proper isoelectric baseline with easily recognizable waveforms.

### 2.4. Electrocardiogram Trace Analysis

ECGs were digitalized immediately after their acquisition The procedure of the ECG interpretation and nomenclature was carried out following standard methods [18]. A board-certified veterinary cardiologist (MJFP) performed a general analysis to ensure the quality of the ECG recording and evaluate cardiac rhythm. Then, two independent observers (LCB and VCG) carried out the manual ECG analysis. Each observer evaluated three representative consecutive beats to measure various ECG variables for each animal; that is, 6 beats per animal were analyzed (588 cardiac beats in total). Then, data obtained for each variable were averaged to have a representative value for each animal. The variables analyzed included:(i)Cardiac rhythm, which was classified as normal sinus rhythm, sinus arrhythmia and pathological arrhythmias.(ii)Heart rate (HR), in beats per minute (bpm), which was calculated by determining the number of QRS complexes in a 3 s interval and multiplying this number by 20 (50 mm/s) or 40 (100 mm/s).(iii)Using the bipolar lead II, the following variables were measured (Figure 1B): amplitude and duration of the P wave, PR interval duration, duration of the QRS complex; amplitude of the R and S waves; QT interval duration; amplitude of the T wave. QT measurements were corrected using the Fridericia and Framingham formula: QTc = QT/RR^1/3^.

Mean electrical axis (MEA) in the frontal plane was calculated by the vector method using the leads I and III, and they were then classified as normal (from −30° to 100°), right axis deviation (RAD, from 100° to 180°), extreme right axis deviation (EAD, from −90° to −180°) and left axis deviation (LAD, from −30° to −90°) [19].

### 2.5. Statistical Analysis

Normal distribution of the values was checked by the Shapiro–Wilk test and multiple comparisons were performed according to the normality of data. When the variable considered was age (i.e., grouping females and males of the same age and comparing different ages), ordinary one-way ANOVA (followed by Tukey multiple comparison test) or Kruskal–Wallis (followed by Dunn’s multiple comparison test) was applied depending on the normal distribution of the data. On the other hand, comparisons between males and females (excluding the age variable) were performed using the *t*-test (unpaired or Mann–Whitney). Spearman’s test was applied for the correlations analyses. Statistical differences were always considered as significant when *p* < 0.05.

## 3. Results

The protocol adopted for the ECG recordings in anesthetized *O. degus* allowed us to obtain tracings of sufficient quality to analyze the HR and cardiac rhythm, as well as evaluate the different waves and intervals, considering a minimum of three consecutive cardiac cycles. However, the movement of the baseline of the ECG tracing was commonly found. The values obtained for each parameter of the ECG when animals were divided regarding age or sex are presented in Table 2 and Table 3. The values obtained for each parameter of the ECG when animals were divided by both age and sex are presented in Table 4 and Table 5.

### 3.1. Heart Rate

The values for the HR of all the animals used in this study ranged from 110 (old males) to 340 (senile females) bpm (Table 2). No significant differences were found when all females and all males were compared (Figure 2A, Table 2). On the contrary, when animals were grouped by age, the HR significantly decreased in the old and senile animals compared with the young ones (Figure 2B, Table 2). When increasing the age, the HR values of the males appeared to be more affected than those from females; even if we detected a decrease in the HR within the aging process in both sexes, this became statistically significant only when young and old females were compared (Figure 2C, Table 4).

The possible correlation between HR and other parameters was evaluated in all the animals (Figure 2D–H). HR showed a significant negative correlation with body weight (Figure 2D). Regarding the duration of the different waves and intervals, HR showed a significant negative correlation with PR duration (Figure 2E). A negative correlation between the QRS complex and HR was found, although it did not reach statistical significance (Figure 2G). No significant relationship was found between HR and QTc (Figure 2F), nor with P wave duration (Figure 2H).

### 3.2. Rhythm

The evaluation of the cardiac rhythm showed that both sex and age influence the appearance of an altered cardiac rhythm (Table 5). In the females, only 5% of the ECG tracings evaluated showed arrhythmia (Table 5). On the contrary, alterations in the rhythm were more predominant in males, and unexpectedly those that were juvenile (Table 5). Figure 3 shows a representative ECG recording of a physiological rhythm, and Figure 4 represents the different pathological rhythms found during aging in the *O. degus*.

### 3.3. P Wave

The duration of the P wave (ms) of all the animals oscillated between 0.005 (young females) and 0.040 (senile females and juvenile males) (Table 3 and Table 6). Females and males exert opposite behavior regarding the duration of the P wave during aging: the duration of the P wave decreases as males get older, whereas it increases with aging in females (Figure 5A).

All the values for the voltage (mV) of the P wave were found between the range 0.006 (young females)–0.083 (senile males) (Table 2 and Table 3). No significant differences were found in the multiple comparison test, neither regarding sex nor age (Figure 5B).

### 3.4. PR Interval

The values of the PR interval (ms) ranged from 0.020 (senile males) to 0.096 (senile females) (Table 3 and Table 6). Significant differences in the PR interval duration were found when males and females were compared: males showed significantly smaller values than females (Table 3). This parameter did not change significantly with aging in the *O. degus* (Table 3, Figure 5C).

### 3.5. QRS Complex

The QRS complex duration (ms) ranged from 0.020 (young females) to 0.075 (senile females) (Table 3 and Table 6). The statistical analysis determined that the effect of “age” was extremely significant. In particular, the duration of the QRS is enlarged during aging in the species *O. degus*: senile animals presented significantly larger QRS complexes compared with the young animals (Table 3). The effect of age on this parameter was more pronounced in the females than in the males: there were no significant differences when aging increased in males, whereas, in the group of females, the duration of the QRS complex in the senile females was significantly higher (increase of 51.51%) and very significantly higher (increase of 61.29%) when compared to juvenile and young females, respectively (Figure 5D).

Finally, the effect of “sex” alone (females vs. males) on the QRS duration was not found to be significant (Table 3).

### 3.6. R Wave (mV)

The amplitude of the R wave (mV) was inside the value range between 0.013 (old females) and 0.525 (juvenile males) (Table 2 and Table 4). The statistical analysis showed no significant differences when the multiple comparison test was performed (Figure 5E). The effect of “age” and “sex” were not significant for explaining the variations observed in the values among the different experimental groups (Table 2).

### 3.7. S Wave (mV)

The values of the S wave voltage (mV) in lead II were between 0.002 (old females) and 0.363 (juvenile males) (Table 2 and Table 4). The variable “sex” exerts a very significant effect on the total variance of this parameter. Independent of the age, the S wave voltage values were higher in males than in females (Table 2, Figure 5F). The aging process did not influence this parameter significantly (Table 2).

### 3.8. T Wave (mV)

The voltage values of the T wave in lead II ranged from 0.006 (young males) to 0.200 mV (juvenile males) (Table 2 and Table 4). The values of the T wave voltage were higher in juvenile animals (both females and males) than in the rest of the groups, although no significant differences were found (Table 2). No sex-associated significant differences were found (Table 2, Figure 5G).

### 3.9. QT Interval Corrected (QTc)

The values of QTc oscillated between 0.0010 (juvenile females) and 0.0044 (senile males) (Table 3 and Table 6). Statistical differences associated with the age of the animals were found both in males and in females when the different groups were compared with the juvenile one, following a similar pattern: (1) QTc of young animals was significantly decreased when it was compared to the juvenile group; (2) QTc of old and senile animals was significantly increased compared with the juvenile group (Figure 5H). Sex differences were statistically significant in the younger ages (juvenile and young): in the juvenile age, the QTc of males was significantly higher compared to females, whereas, in the young age, the QTc of males was significantly reduced compared to females (Figure 5H).

### 3.10. Electrical Axis

The MEA in the frontal plane of all the animals ranged from −138° (juvenile males) to +180° (senile females) (Table 2 and Table 4). The widest range of the MEA was found in juvenile males (−138.0° to +169.0°), whereas the narrowest one corresponded to young males (−67.0° to +20.5°). When the variable sex was studied, it was found that males had significant lower mean values of the MEA (Figure 6A). Regarding the variable age, the MEA values did not show significant statistical differences (Figure 6B).

Three representative ECGs with a normal electrical axis and left and right deviations are shown in Figure 7.

Animals were classified according to their electrical axis values, and the proportion (as percentage, %) of animals per group was calculated (Figure 6C). The distribution of males and females showed clear differences. Males showed different behavior (Figure 6D). In contrast, in the females, a normal electrical axis predominated, although deviations in the axis appeared as the age increases (Figure 6D). In the young male group, more than 50% of the animals presented deviations from the axis: 3/12 showed LAD, 2/12 showed EAD and 2/12 showed RAD. In adult males, only LAD was found, but it was present in more than 50% of the animals of the group. Surprisingly, in the old and senile group of males, 85% and 81.82% of the animals, respectively, showed a normal electrical axis, and none of them showed an extreme deviation from the axis.

### 3.11. Heart/Body Weight Ratio Is Increased in the Aged O. degus

We studied whether heart weight is influenced by sex and age in the *O. degus* by calculating the heart weight: body weight ratio (expressed as %). There were no significant differences between females and males regarding this parameter (Figure 8A). However, with aging, the heart size increases significantly, mainly in the old and senile animals compared to both juvenile and young animals (Figure 8B).

Additionally, a possible correlation between heart weight and the ECG parameters measured was evaluated (Appendix A). Among them, a very significant negative correlation between R wave amplitude and the heart weight was found (Figure 8C).

## 4. Discussion

This is the first study that comprehensively reports the ECG measurements in the *O. degus*. Animals were divided into eight groups depending on age and sex. The results show that age is the most determinant variable that influences the amplitude and duration of the waves and segments in the ECG recording, although particular sex-related differences were also found. In particular, the QRS complex and the corrected QT segment were the most age-dependent measurements. On the contrary, electrical axis values and the outcome of different types of arrhythmias are more affected by the sex of the animals. In particular, young males presented more variations in the ECG than older ones, specifically in relation to R and S waves, QTc, the incidence and severity of arrhythmias and electrical axis deviation. Arrhythmias and mean electrical axis deviations in young males can be explained by the confluence of an increased R wave duration and T wave amplitude, and a decreased S wave, as well as in variations in QTc. To the author’s knowledge, this is the first study that provides reference values for ECG parameters in the species *O. degus*, acknowledging the effect of age and sex.

Cardiovascular diseases (CVDs) are the diseases with the biggest impact and the first cause of death worldwide [20]. The electric activity of the heart changes throughout life and it has been found to be particularly altered during aging [21]. Several studies have found impairment in the myocites’ cell–cell communication, molecular and membrane properties changes or remodeling of the extracellular matrix, among others [22]. These cellular and molecular variations have structural and functional consequences, such as an altered electric activity of the heart, evidenced by ECG recordings. It is known that aging is an important risk factor for developing CVD in both sexes in humans, but increasing evidence suggests that there are sex differences regarding the cardiac changes that happen during the aging process, as well as in the onset and progression of cardiac diseases [23,24]. For example, atrial hypertrophy and fibrosis are more prevalent in men but have a worse outcome in women. On the other side, left ventricular fibrosis, alterations in calcium metabolism and vascular stiffness are more prevalent in women but show worse outcomes in men [25]. However, the current strategies in the clinical procedures applied to CVD have been stablished from information inferred from studies performed mostly in men because, historically, clinical trials have been made considering the male sex [23,25,26,27].

Despite the fact that age and sex strongly influence cardiac parameters, few studies have incorporated both variables simultaneously. In line with this, the diurnal rodent *O. degus* represents an interesting alternative as an experimental model for age-related pathologies since it spontaneously develops some age-associated traits, including atherosclerosis [15]. However, the cardiac parameters of this species have not been determined yet. Thus, in the present work, we studied the ECG of the *O. degus* with aging and with a sex perspective under anesthesia conditions.

In these animals, the HR values oscillate between 110 and 340 bpm. The HR values reach their maximum value in young animals (1 year old) and decrease with an advancing age. We found that this decrease was particularly significant in old males compared to young. These results are in line with what has been established for humans, mice and rats [28,29,30,31]. On the contrary, no significant sex differences were found in the mean HR value in the *O. degus*.

Heart rhythm analysis revealed that abnormal rhythms are more frequent in males than in females, and that they occur mainly in younger groups (juvenile and young males). It is known that sedation or anesthesia with α-2 agonists in small animals induce arrythmias associated with an increased parasympathetic tone, including a sinus arrhythmia further characterized as a sinus bigeminy [32]. Sex differences have also been reported in humans in cardiac effects in response to different anesthetics [33]. Therefore, the use of the *O. degus* in cardiac research from the pharmacokinetics perspective might be useful for shedding light on sexual dimorphism in the response to anesthesia.

The P waves reflect atrial depolarization, and it has been established that their shape and duration have clinical importance in humans [11]. In general, an increase in P wave duration may be associated with an increased susceptibility to arrhythmias. In the *O. degus*, we observed the opposite behavior when we compared males and females: in females, the duration increases with age, whereas, in males, it decreases. As for the amplitude of the P wave, males showed homogeneous values with age. In contrast, females showed an increase in the value of old and senile ages. Previous studies have suggested that an increased P wave duration is indicative of slower atrial conduction, possibly due to an altered conductive function of myocytes by modifying the expression of key proteins in the myocyte-to-myocyte connection–communication [25,31].

Regarding the duration of the PR interval, we observed an increase in this parameter as the animals age, in agreement with what is described for humans and mice [25,34]. It is worth noting that the highest values of the PR interval duration in females were found in the young and the senile groups, whereas, in males, its prolongation was linear with age. Atrio-ventricular blocks of first grade have been related to an increase in the PR interval in humans during aging [28,35].

The QRS complex, which corresponds to the depolarization of the ventricles, gives important information about the electrical function of the heart and is one of the most affected parameters during aging in humans and rodents [11,34]. We found that its duration increased significantly with age in females, whereas, in males, this increase was smaller. No differences were found when comparing males and females of the same age, although the values for males had a tendency to be higher at all ages except for the senile age. These results are consistent with human studies since it has been shown that the duration and voltage of this complex increases with age and that values for men are higher than those for women [27]. An increase in the duration of this complex is associated with alterations in ventricular rhythms and ventricular conduction, which, in turn, are related to heart failure and myocardial ischemia [11].

Within the QRS complex, the analysis of the R and S wave amplitude also showed a sex-dependent behavior. In females, the R wave showed a decrease from young to old ages but increased in senile females. On the contrary, the trend with age in males was found to be very variable, and adult males showed a large decrease in the voltage of this wave. In the S wave amplitude, the values for females remained similar at juvenile, young and old ages, although they increased slightly in the senile animals. In males, the values were higher at all ages and significantly higher in young males. Alterations in S and R wave amplitudes suggest alterations in the depolarization of the ventricles in young males.

In relation to the T wave amplitude, the maximum values were found in the juvenile group of both sexes. Then, these values decreased in the young groups and tended to increase in the old and senile females, whereas they tended to decrease in the old and senile males. Knowing that the T wave corresponds to ventricular repolarization, abnormal values for this wave may suggest cardiac pathology, such as arrhythmias [11,28]. Higher values of T wave voltage may indicate hyperkalemia, whereas lower values are indicative of hypokalemia [11,27,36].

Ventricular repolarization is measured as the duration of the QT interval in the ECG. In the *O. degus*, the duration of this segment was prolonged in the oldest groups both in males and in females. The highest sex differences were found in the younger ages. The prolongation of the QT interval has been closely related to the toxicity of exogenous compounds and can predispose patients to myocardial infarction, channelopathies, ischemia or arrhythmias in both humans and laboratory animals [11,37]. Our data highlight that QTc is strongly influenced by the age and the sex in this species, and future studies (both in vivo and postmortem) would be very promising for shedding light on these differences.

The study of the electrical axis can provide important information for finding underlying pathologic states and a better characterization of the disease [19]. Axis deviation is more frequent with aging. This behavior was observed in the females of this study. Surprisingly, the juvenile and young males were the ones that presented the greatest axis deviation, which decreased with age (similar to arrhythmias). The most frequent axis deviation in males was LAD, which has also been described as the most frequent in humans, and which has been related to changes in the position of the heart and anterior fascicle fibrosis [31].

Aged animals (independent from the sex) showed a greater heart weight/body weight ratio: those of the old and senile animals were 1.64-fold and 1.75-fold greater, respectively, compared to the juvenile ones. This cardiac hypertrophy was negatively correlated (very significantly) with the R wave voltage. Several lines of evidence have pointed out that an increase in the heart weight can be both a physiological response and a pathological situation [38,39]. Thus, future studies might focus on a deeper cellular and molecular characterization of heart hypertrophy in the *O. degus*.

Rabbit, dog and pig ECGs are more similar to human ECGs. However, mice are the most used model for studying cardiac electrophysiology because rodents have several advantages compared to other models, such as a smaller size, easier breeding and availability of laboratory facilities, easier handling, lower maintenance costs and ethical implications [40]. The murine ECG is characterized by ECG elements that are not present in the human ECG, including the presence of the J wave, the inverted T wave and the absence of the ST segment [28,40,41]. In this sense, the ECG of the *O. degus* was found to be more similar to the human one due to the shape of the T wave, the presence of an ST segment, a more pronounced Q wave and a J wave absence. Previous works have shown that the differences observed between the human ECG and the preclinical models come from the distribution and diversity of the activation pattern of ion channels [41]; therefore, future studies on the ion channels of the *O. degus* heart would be useful for the characterization of the cardiac system of this emerging model. Therefore, considering that the *O. degus* belongs to the rodent family, the advantages of using mice or rats are maintained and, at the same time, the ECG dissimilarities are suppressed.

Altogether, these findings provide reference values for the ECG recording in the *O. degus*, providing evidence that this model may represent a valuable tool for studying cardiac function in the context of aging. The variability observed in the data provides the opportunity to understand the individual susceptibility to different pathological conditions. In addition, this study reinforces the need for including both sexes in preclinical research since significant differences were found between males and females in the different ages considered. These results might serve as the starting point for future studies aiming at deciphering the aging heart mechanisms from the perspective of sex differences and individual susceptibility.

### Limitations

In this study, some limitations can be considered. The main limitation was that the ECGs were recorded with the animals under anesthesia, and individual variations in heart rate and rhythm are known to occur with the anesthetic agents used in this study [42,43,44]. In this study, we cannot know if medetomidine has contributed to the appearance of arrhythmias or not since the electrocardiograms were not performed without anesthesia in the same animals. However, anesthesia is necessary and recommended for most clinical procedures in small animals, and the combination of dexmedetomidine and ketamine is commonly used and has shown to be effective. In addition, the sample size is considered as sufficient for establishing reference values and finding significant large effects with aging in the *O. degus*. However, in order to observe significant small effects, the population size should be increased (especially in the younger groups).

## 5. Conclusions

In this study, we characterized for the first time the electrocardiographic parameters in the *Octodon degus* during the aging process, considering both females and males. The data obtained showed spontaneous cardiac alterations with age that differ in females and males. Therefore, we suggest that degus may act as an attractive natural model for cardiovascular research since it shares features relevant to human aging and presents signs of sex-specific alterations.

## Figures and Tables

**Figure 1 biology-12-00747-f001:**
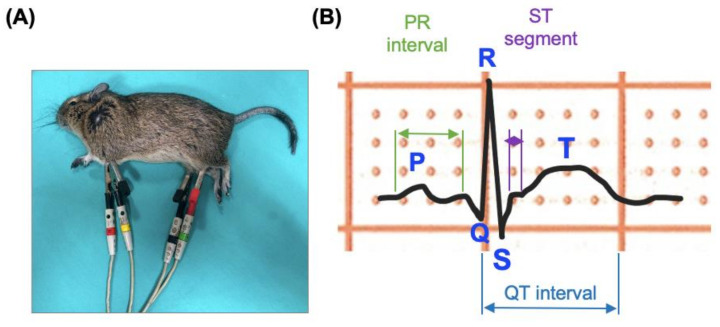
Experimental setting of the study. (**A**) Photograph of an anesthetized *O. degus* while the ECG was recorded; (**B**) Schematic representation of the waves and intervals measured in the lead II of the *O. degus* ECG recording. Blue letters indicate that the voltages of these waves were evaluated, while duration of the different intervals and segments measured are represented by horizontal delimitated arrows.

**Figure 2 biology-12-00747-f002:**
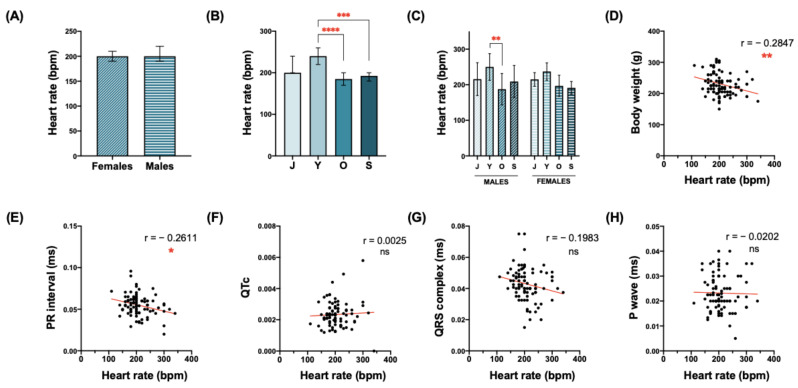
Heart rate values in the *O. degus*. Graphical representation of the heart rate regarding sex (**A**), age (**B**) or both variables simultaneously (**C**). (**D**–**H**) Correlations between heart rate values and: body weight (**D**), PR interval duration (**E**), QT interval corrected (**F**), QRS complex duration (**G**) and P wave duration (**H**). Statistical analysis used to study the differences between males and females in the heart rate was performed by the Mann–Whitney test. Statistical analysis used to study the effect of age on heart rate was performed by Kruskal–Wallis + Dunn’s multiple comparison test. Data are expressed as mean ± standard deviation in the column graphs. Data are presented as individual points in the correlation graphs, with the red line indicating linear regression. Number of XY pairs = 87. In both types of graphs, asterisks indicate statistical differences: * *p* < 0.05, ** *p* < 0.01, *** *p* < 0.001, **** *p* < 0.0001. Abbreviations: bpm = beats per minute; QTc = QT interval corrected; ms = milliseconds; r = Spearman’s coefficient; ns = not significant; J = juvenile; Y = young; O = old; S = senile.

**Figure 3 biology-12-00747-f003:**
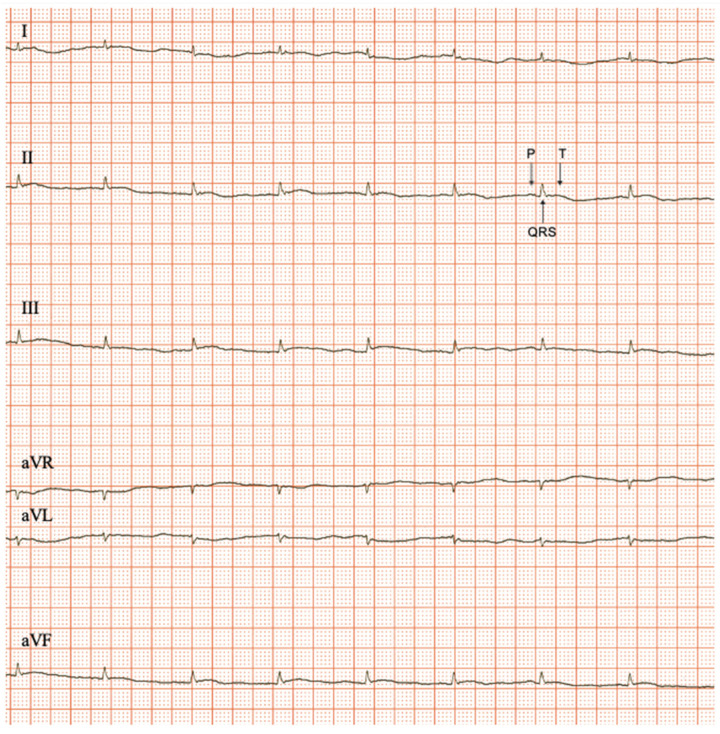
Representative six-lead electrocardiogram of sinus rhythm; 50 mm/s and 20 mm/mV. From top to bottom: leads I, II and III (Einthoven’s leads), leads aVR, aVF and aVL. (Goldberger leads).

**Figure 4 biology-12-00747-f004:**
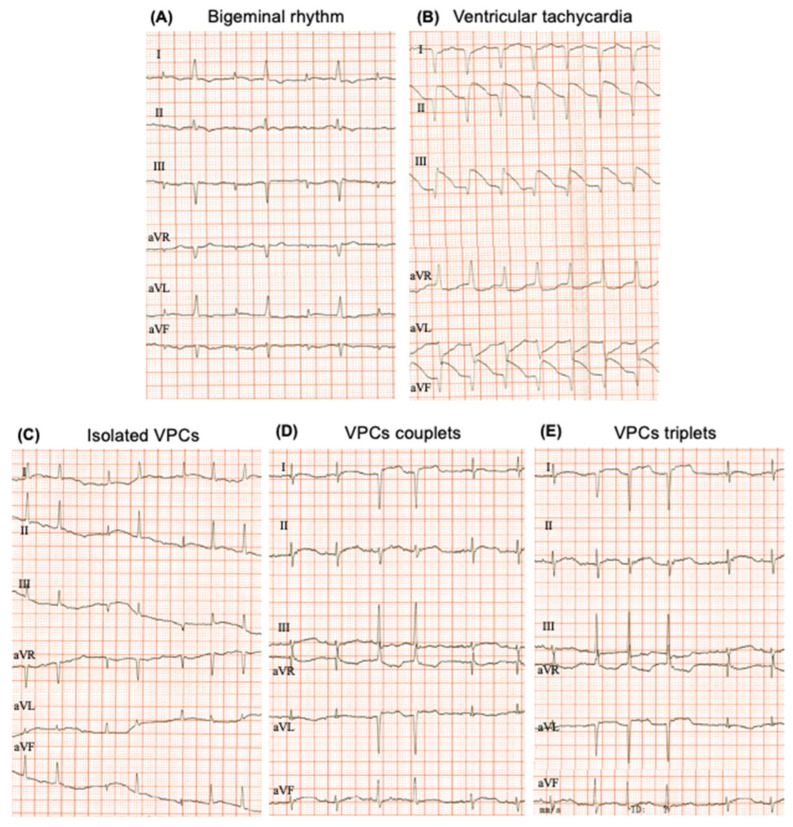
Representative electrocardiograms of the main different pathological ECG recordings found in the study: (**A**) bigeminal rhythm; (**B**) ventricular tachycardia; (**C**) isolated ventricular premature complexes (VPCs) (**D**) VPCs couplets and (**E**) triplets; 50 mm/s and 20 mm/mV. From top to bottom: leads I, II and III (Einthoven’s leads), leads aVR, aVF and aVL. (Goldberger leads).

**Figure 5 biology-12-00747-f005:**
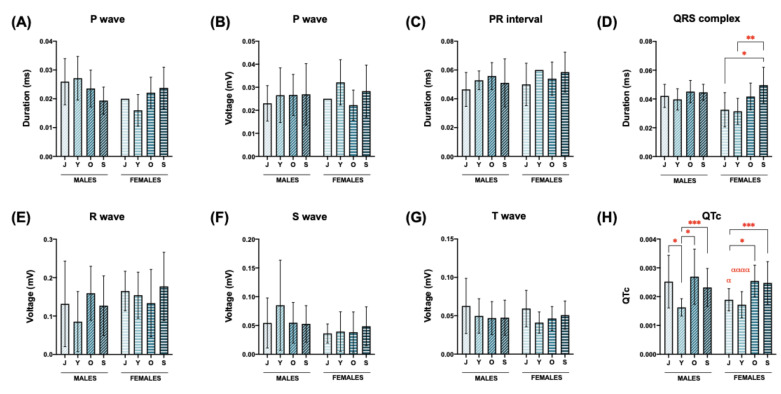
Effect of sex and age on the electrocardiographic parameters in the *O. degus*. (**A**) P wave duration (ms); (**B**) P wave voltage (mV); (**C**) PR interval duration (ms); (**D**) QRS complex duration (ms); (**E**) R wave voltage (mV); (**F**) S wave voltage (mV); (**G**) T wave voltage (mv); (**H**) corrected QT interval. Statistical analysis used to study the effect of age on the ECG parameters was performed by Kruskal–Wallis + Dunn’s multiple comparison test. Statistical analysis used to study the sex-associated differences in the different ECG parameters was performed by the unpaired *t*-test. Data are expressed as mean ± standard deviation. Asterisks indicate statistical differences comparing animals of the same sex and different age: * *p* < 0.05, ** *p* < 0.01, *** *p* < 0.001. Alpha character (α) indicates statistical differences when males and females of the same age are compared (α *p* < 0.05, αααα *p* < 0.0001). Abbreviations: ms = milliseconds; J = juvenile; Y = young; O = old; S = senile.

**Figure 6 biology-12-00747-f006:**
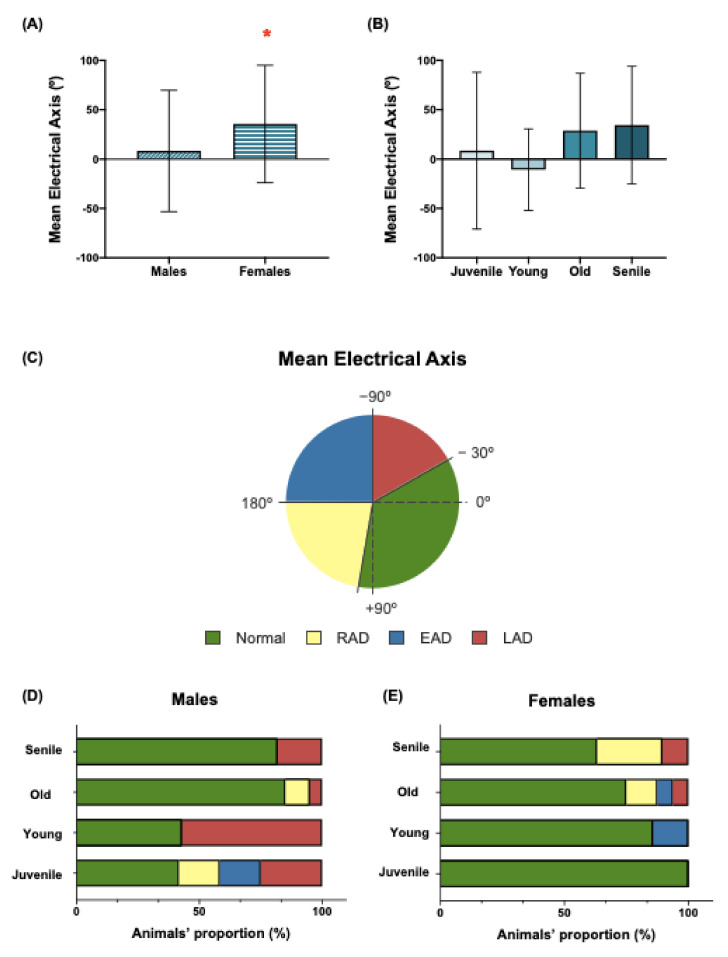
Electrical axis in the *O. degus* females and males during aging. (**A**) Graphical representation of the electrical axis values in the *O. degus* regarding sex and (**B**) age. (**C**) Classification of the electrical axis: normal, right axis deviation (RAD), extreme axis deviation (EAD) and left axis deviation (LAD). According to this classification, the electrical axis of the animals was calculated to express the proportion of animals with each axis deviation in males (**D**) and females (**E**). Statistical analysis used to study the sex-associated differences in the different ECG parameters was performed by the unpaired *t*-test. Statistical analysis used to study the effect of age on the ECG parameters was performed by ordinary one-way ANOVA + Tukey multiple comparison test. Asterisks indicate statistical difference between males and females: * *p* < 0.05. Data are expressed as mean ± SD in the column graphs.

**Figure 7 biology-12-00747-f007:**
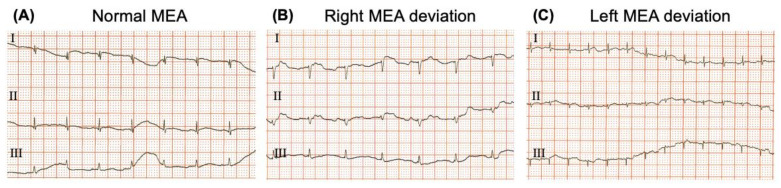
Representative electrocardiograms of the main different electrical axis deviations identified in the study: (**A**) normal axis deviation; (**B**) right axis deviation: and (**C**) left axis deviation; 50 mm/s and 20 mm/mV.

**Figure 8 biology-12-00747-f008:**
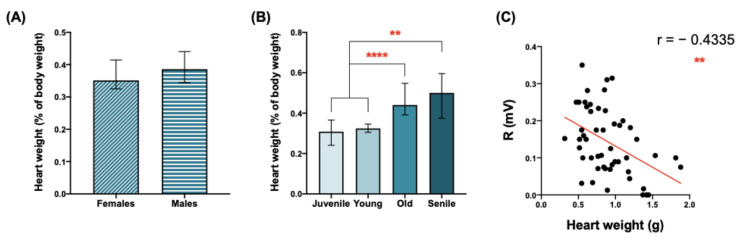
Heart weight changes with aging. Heart weight (% with respect to the total body weight) regarding sex (**A**) and age (**B**). Statistical analysis used to study the differences between males and females in the heart rate was performed by the Mann–Whitney test. Statistical analysis used to study the effect of age on heart rate was performed by ordinary one-way ANOVA + Tukey multiple comparison test. Data are expressed as mean ± standard deviation. (**C**) Correlations between and R voltage and heart weight. Data are expressed as individual values in the correlations, with red line indicating linear regression. Number of XY pairs = 55. Asterisks indicate the significance of the statistical differences: ** *p* < 0.01, **** *p* < 0.0001. Abbreviations: r = Spearman’s coefficient.

**Table 1 biology-12-00747-t001:** Number of animals (n) included in each experimental group and their weights (g). Animals were divided according to sex (males and females) and age (juvenile, young, old and senile). Data are expressed as mean ± standard deviation.

	Juvenile (6 Months Old)	Young (1 Year Old)	Old (4–5 Years Old)	Senile (6–7 Years Old)
Males	n = 13	n = 7	n = 20	n = 12
Weight (g)	194.6 ± 14.4	220.7 ± 18.5	264.3 ± 33.5	246.7 ± 33.3
Females	n = 4	n = 7	n = 16	n = 19
Weight (g)	202.5 ± 15.5	205.0 ± 11.4	226.8 ± 18.8	221.5 ± 33.9

**Table 2 biology-12-00747-t002:** Reference values for the voltage of the ECG waves, heart rate (HR) and mean electrical axis (MEA). The data presented correspond to: (A) the values of all the animals analyzed (no subdivisions) and divided either by sex (B) or age (C). Mean ± standard deviation (SD), median, first and third quartiles (Q1, Q3) and minimum–maximum values are reported. bpm = beats per minute.

Groups	HR (bpm)	P (mV)	R (mV)	S (mV)	T (mV)	MEA (°)
**ALL ANIMALS TOGETHER**						
Mean ± SD	207.2 ± 41.3	0.023 ± 0.007	0.168 ± 0.112	0.065 ± 0.063	0.053 ± 0.029	21.4 ± 61.7
Median	200.0	0.025	0.150	0.044	0.050	19.3
Q1 | Q3	180.0 | 230.0	0.019 | 0.032	0.090 | 0.217	0.025 | 0.075	0.038 | 0.063	−17.3 | 60.0
Min–Max	110.0–340.0	0.006–0.083	0.013–0.525	0.002–0.363	0.006–0.200	−138.0–180.0
**BY SEX**						
**Males**						
Mean ± SD	209.0 ± 47.0	0.026 ± 0.010	0.150 ± 0.078	0.057 ± 0.039	0.053 ± 0.026	8.3 ± 61.4
Median	200.0	0.025	0.146	0.044	0.051	0.0
Q1 | Q3	180.0 | 240.0	0.019 | 0.032	0.088 | 0.210	0.023 | 0.085	0.036 | 0.075	−29.0 | 39.0
Min–Max	110.0–320.0	0.013–0.050	0.019–0.525	0.013–0.363	0.006–0.129	−138.0–169.0
**Females**						
Mean ± SD	203.0 ± 28.3	0.027 ± 0.010	0.165 ± 0.100	0.043 ± 0.032	0.050 ± 0.021	35.6 ± 59.5
Median	200.0	0.025	0.152	0.032	0.050	39.0
Q1 | Q3	180.0 | 220.0	0.020 | 0.032	0.095 | 0.230	0.025 | 0.058	0.038 | 0.061	2.63 | 66.9
Min–Max	160.0–270.0	0.006–0.050	0.013–0.504	0.002–0.125	0.013–0.117	−128.0–180.0
**BY AGE**						
**Juvenile**						
Mean ± SD	216.0 ± 40.5	0.025 ± 0.009	0.183 ± 0.133	0.067 ± 0.058	0.071 ± 0.047	8.5 ± 79.4
Median	200.0	0.025	0.168	0.044	0.072	−5.75
Q1 | Q3	200.0 | 235.0	0.019 | 0.0303	0.085 | 0.231	0.025 | 0.103	0.032 | 0.087	−46.9 | 70.5
Min–Max	140.0–300.0	0.013–0.050	0.025–0.525	0.015–0.363	0.008–0.200	−138.0–169.0
**Young**						
Mean ± SD	244.0 ± 31.0	0.030 ± 0.011	0.157 ± 0.087	0.049 ± 0.044	0.046 ± 0.075	−10.7 ± 41.3
Median	240.0	0.025	0.138	0.029	0.050	−1.25
Q1 | Q3	220.0 | 260.0	0.025 | 0.038	0.087 | 0.198	0.019 | 0.053	0.035 | 0.063	−45.5 | 21.0
Min–Max	200.0–320.0	0.013–0.050	0.063–0.363	0.013–0.163	0.006–0.019	−90.0–46.0
**Old**						
Mean ± SD	187.0 ± 30.2	0.025 ± 0.010	0.148 ± 0.078	0.049 ± 0.035	0.049 ± 0.023	28.8 ± 58.1
Median	185.0	0.025	0.138	0.044	0.044	23.3
Q1 | Q3	170.0 | 210.0	0.015 | 0.031	0.091 | 0.198	0.019 | 0.077	0.038 | 0.062	−11.0 | 63.5
Min–Max	110.0–270.0	0.006–0.044	0.013–0.313	0.002–0.138	0.007–0.113	−128.0–163.0
**Senile**						
Mean ± SD	198.0 ± 31.5	0.027 ± 0.011	0.019 ± 0.504	0.051 ± 0.033	0.051 ± 0.018	34.4 ± 59.6
Median	193.0	0.026	0.163	0.040	0.050	27.3
Q1 | Q3	180.0 | 200.0	0.020 | 0.037	0.085 | 0.226	0.029 | 0.065	0.038 | 0.057	−2.88 | 70.5
Min–Max	160.0–300.0	0.012–0.083	0.182–0.116	0.001–0.125	0.019–0.100	−67.0–180.0

**Table 3 biology-12-00747-t003:** Reference values for the amplitude of the ECG waves and intervals. The data presented correspond to: (A) the values of all the animals analyzed (no subdivisions), divided either by sex (B) or age (C). Mean ± standard deviation (SD), median, first and third quartiles (Q1, Q3) and minimum–maximum values are reported. Asterisks indicate statistical differences: * *p* < 0.05; ** *p* < 0.01; **** *p* <0.0001.

Groups	P (ms)	PR Interval (ms)	QRS (ms)	QTc
**ALL ANIMALS TOGETHER**				
Mean ± SD	0.023 ± 0.007	0.055 ± 0.013	0.044 ± 0.010	0.155 ± 0.029
Median	0.022	0.055	0.043	0.0023
Q1 | Q3	0.005–0.040	0.047 | 0.063	0.038 | 0.050	0.0019 | 0.0028
Min–Max	0.005–0.040	0.020–0.096	0.020–0.075	0.0010–0.0044
**BY SEX**				
**Males**				
Mean ± SD	0.024 ± 0.007	0.052 ± 0.012	0.044 ± 0.007	0.158 ± 0.029
		* *vs females*		
Median	0.022	0.053	0.045	0.0023
Q1 | Q3	0.018 | 0.030	0.047 | 0.060	0.040 | 0.050	0.0019 | 0.0028
Min–Max	0.013–0.040	0.020–0.073	0.025–0.056	0.0013–0.0050
**Females**				
Mean ± SD	0.022 ± 0.007	0.058 ± 0.013	0.043 ± 0.013	0.151 ± 0.026
Median	0.022	0.060	0.040	0.0023
Q1 | Q3	0.018 | 0.026	0.048 | 0.063	0.035 | 0.049	0.0019 | 0.0028
Min–Max	0.005–0.040	0.034–0.096	0.020–0.075	0.0010–0.0044
BY AGE				
**Juvenile**				
Mean ± SD	0.025 ± 0.007	0.048 ± 0.012	0.040 ± 0.010	0.0022 ± 0.0005
Median	0.025	0.049	0.040	0.0021
Q1 | Q3	0.019 | 0.030	0.036 | 0.060	0.040 | 0.047	0.0017 | 0.0024
Min–Max	0.015–0.040	0.030–0.065	0.020–0.056	0.0015–0.0034
**Young**				
Mean ± SD	0.022 ± 0.010	0.058 ± 0.008	0.036 ± 0.009	0.0017 ± 0.0004
Median	0.020	0.060	0.039	0.0017
Q1 | Q3	0.014 | 0.030	0.050 | 0.060	0.030 | 0.040	0.0014 | 0.0020
Min–Max	0.005–0.035	0.045–0.075	0.020–0.053	0.0010–0.0024
**Old**				
Mean ± SD	0.023 ± 0.006	0.057 ± 0.012	0.044 ± 0.009	0.0025 ± 0.0006
				**** *vs. young*
Median	0.023	0.055	0.045	0.0025
Q1 | Q3	0.020 | 0.028	0.049 | 0.065	0.036 | 0.050	0.0021 | 0.0030
Min–Max	0.013–0.035	0.035–0.090	0.025–0.058	0.0013–0.0036
**Senile**				
Mean ± SD	0.023 ± 0.007	0.056 ± 0.015	0.049 ± 0.012	0.0024 ± 0.0007
			** *vs. young*	** *vs. young*
Median	0.022	0.058	0.049	0.0024
Q1 | Q3	0.017 | 0.027	0.048 | 0.064	0.041 | 0.053	0.0021 | 0.0028
Min–Max	0.013–0.040	0.020–0.096	0.026–0.075	0.0012–0.0044

**Table 4 biology-12-00747-t004:** Reference values for the voltage of the ECG waves, heart rate (HR) and mean electrical axis (MEA) of the animals divided by sex and age. Mean ± standard deviation (SD), median, first and third quartiles (Q1, Q3) and minimum–maximum values are reported. bpm = beats per minute.

Groups	HR (bpm)	P (mV)	R (mV)	S (mV)	T (mV)	MEA (°)
**MALES**						
**Juvenile**						
Mean ± SD	216.0 ± 46.2	0.023 ± 0.007	0.197 ± 0.149	0.109 ± 0.108	0.074 ± 0.052	−2.0 ± 86.9
Median	200.0	0.022	0.100	0.0438	0.069	−11.5
Q1 | Q3	193.0 | 235.0	0.019 | 0.030	0.031 | 0.225	0.017 | 0.116	0.031 | 0.088	−48.0 | 56.3
Min–Max	140.0–300.0	0.013–0.038	0.031–0.525	0.015–0.363	0.008–0.200	−138.0–169.0
**Young**						
Mean ± SD	250.0 ± 37.4	0.027 ± 0.012	0.120 ± 0.064	0.111 ± 0.098	0.050 ± 0.023	−25.6 ± 32.2
Median	240.0	0.025	0.075	0.0878	0.054	-38.0
Q1 | Q3	230.0 | 270.0	0.019 | 0.032	0.000 | 0.188	0.021 | 0.196	0.038 | 0.063	−47.0 | 9.0
Min–Max	200.0–320.0	0.013–0.050	0.063–0.192	0.013–0.263	0.006–0.075	−67.0–20.5
**Old**						
Mean ± SD	190.0 ± 43.8	0.026 ± 0.009	0.159 ± 0.070	0.062 ± 0.046	0.050 ± 0.026	25.7 ± 55.2
Median	180.0	0.025	0.144	0.0542	0.044	17.0
Q1 | Q3	163.0 | 200.0	0.022 | 0.033	0.100 | 0.189	0.019 | 0.081	0.042 | 0.063	−14.8 | 66.1
Min–Max	110.0–300.0	0.013–0.044	0.081–0.313	0.013–0.188	0.007–0.113	−76.0–163.0
**Senile**						
Mean ± SD	209.0 ± 45.0	0.032 ± 0.021	0.166 ± 0.125	0.053 ± 0.032	0.052 ± 0.018	9.4 ± 47.5
Median	200.0	0.022	0.122	0.044	0.044	9.0
Q1 | Q3	180.0 | 240.0	0.016 | 0.057	0.066 | 0.116	0.020 | 0.072	0.035 | 0.059	−11.5 | 37.0
Min–Max	160.0–300.0	0.013–0.083	0.019–0.442	0.013–0.117	0.035–0.082	−67.0–99.0
**FEMALES**						
**Juvenile**						
Mean ± SD	215.0 ± 19.1	0.025 ± 0.000	0.15 ± 0.086	0.025 ± 0.000	0.059 ± 0.024	39.8 ± 45.7
Median	210.0	0.025	0.168	0.030	0.069	52.0
Q1 | Q3	200.0 | 235.0	0.025 | 0.025	0.115 | 0.213	0.025 | 0.054	0.034 | 0.075	−8.3 | 75.5
Min–Max	200.0–240.0	0.025–0.025	0.025–0.225	0.025–0.025	0.025–0.0750	−21.0–76.0
**Young**						
Mean ± SD	200.0 ± 260.0	0.032 ± 0.010	0.184 ± 0.096	0.038 ± 0.019	0.041 ± 0.014	4.1 ± 46.3
Median	240.0	0.025	0.138	0.025	0.038	21.0
Q1 | Q3	220.0 | 260.0	0.025 | 0.038	0.100 | 0.213	0.013 | 0.05	0.025 | 0.050	−19.0 | 37.0
Min–Max	237.0–24.3	0.025–0.050	0.100–0.363	0.013–0.063	0.025–0.063	−90.0–46.0
**Old**						
Mean ± SD	197.0 ± 29.8	0.022 ± 0.010	0.134 ± 0.088	0.041 ± 0.035	0.047 ± 0.019	32.6 ± 63.2
Median	195.0	0.019	0.114	0.025	0.045	43.0
Q1 | Q3	173.0 | 210.0	0.016 | 0.029	0.065 | 0.225	0.013 | 0.055	0.038 | 0.056	−7.5 | 63.5
Min–Max	160.0–270.0	0.006–0.044	0.013–0.283	0.002–0.119	0.013–0.083	−128.0–147.0
**Senile**						
Mean ± SD	191.0 ± 17.8	0.027 ± 0.010	0.212 ± 0.147	0.052 ± 0.034	0.050 ± 0.019	48.9 ± 62.1
Median	193.0	0.027	0.165	0.036	0.050	50.5
Q1 | Q3	180.0 | 200.0	0.021 | 0.033	0.113 | 0.272	0.027 | 0.068	0.039 | 0.056	3.5 | 102.0
Min–Max	160.0–230.0	0.012–0.044	0.033–0.033	0.007–0.125	0.019–0.100	−51.0–180.0

**Table 5 biology-12-00747-t005:** Cardiac rhythm in *O. degus* and incidence of cardiac rhythm alterations. The proportion of animals, expressed as percentage (%) from the total of individuals of the corresponding group, was quantified according to sinus rhythm and different types of arrhythmias (bigeminal rhythm, ventricular premature complexes and ventricular tachycardia). Ventricular premature complexes were subdivided into frequent or isolated, couplets or triplets.

	Sinus Rhythm (%)	Type of Rhythm			
		Bigeminal Rhythm (%)	Ventricular Premature Complexes (VPC)		Ventricular Tachycardia
			Frequent VPC (%)	Isolated VPC, Couplets or Triplets (%)	
**Males**					
Juvenile	37.50	50.00	-	-	12.50
Young	66.67	33.33	-	-	-
Old	100.00	-	-	-	-
Senile	66.67	-	25.00	8.33	-
**Females**					
Juvenile	100.00	-	-	-	-
Young	85.71	-	-	14.29	-
Old	93.75	-	6.25	-	-
Senile	100.00	-	-	-	-

**Table 6 biology-12-00747-t006:** Reference values for the amplitude of the ECG waves and intervals of the animals divided by sex and age. Mean ± standard deviation (SD), median, first and third quartiles (Q1, Q3) and minimum–maximum values are reported.

Groups	P (ms)	PR Interval (ms)	QRS (ms)	QTc
**MALES**				
**Juvenile**				
Mean ± SD	0.026 ± 0.008	0.047 ± 0.012	0.042 ± 0.008	0.0025 ± 0.0009
Median	0.023	0.049	0.041	0.0023
Q1 | Q3	0.018 | 0.034	0.034 | 0.057	0.040 | 0.048	0.0020 | 0.0029
Min−Max	0.015−0.040	0.030−0.063	0.025−0.056	0.0016−0.0050
**Young**				
Mean ± SD	0.027 ± 0.008	0.053 ± 0.007	0.040 ± 0.007	0.00163 ± 0.0003
Median	0.030	0.050	0.040	0.0015
Q1 | Q3	0.020 | 0.035	0.048 | 0.060	0.038 | 0.043	0.0014 | 0.0020
Min−Max	0.015−0.035	0.045−0.063	0.028−0.053	0.0013−0.0021
**Old**				
Mean ± SD	0.024 ± 0.006	0.057 ± 0.009	0.045 ± 0.008	0.0025 ± 0.00064
Median	0.021	0.055	0.048	0.0025
Q1 | Q3	0.020 | 0.029	0.049 | 0.062	0.041 | 0.050	0.0021 | 0.0031
Min−Max	0.014−0.035	0.043−0.073	0.027−0.055	0.0013−0.0035
**Senile**				
Mean ± SD	0.019 ± 0.005	0.051 ± 0.017	0.045 ± 0.010	0.0023 ± 0.0007
Median	0.018	0.052	0.044	0.0023
Q1 | Q3	0.016 | 0.022	0.038 | 0.068	0.040 | 0.051	0.0017 | 0.0026
Min−Max	0.013−0.028	0.020−0.071	0.038−0.052	0.0013−0.0036
**FEMALES**				
**Juvenile**				
Mean ± SD	0.020 ± 0.000	0.050 ± 0.015	0.040 ± 0.000	0.0018 ± 0.0004
Median	0.020	0.050	0.038	0.0019
Q1 | Q3	0.020 | 0.020	0.036 | 0.064	0.020 | 0.040	0.0015 | 0.0021
Min−Max	0.020−0.020	0.035−0.065	0.015−0.040	0.0010−0.0036
**Young**				
Mean ± SD	0.016 ± 0.008	0.060 ± 0.000	0.031 ± 0.009	0.0026 ± 0.0006
Median	0.020	0.060	0.030	0.0024
Q1 | Q3	0.010 | 0.020	0.060 | 0.060	0.020 | 0.040	0.0020 | 0.0030
Min−Max	0.005−0.030	0.040−0.040	0.020−0.040	0.0019−0.0024
**Old**				
Mean ± SD	0.022 ± 0.005	0.058 ± 0.015	0.042 ± 0.009	0.0025 ± 0.0007
Median	0.023	0.055	0.041	0.0024
Q1 | Q3	0.018 | 0.025	0.046 | 0.063	0.035 | 0.048	0.0021 | 0.0028
Min−Max	0.013−0.032	0.035−0.090	0.025−0.058	0.0012−0.0036
**Senile**				
Mean ± SD	0.025 ± 0.008	0.059 ± 0.014	0.050 ± 0.012	0.0572 ± 0.0074
Median	0.023	0.059	0.049	0.1500
Q1 | Q3	0.018 | 0.033	0.048 | 0.064	0.040 | 0.055	0.0021 | 0.166
Min−Max	0.013−0.040	0.034−0.096	0.026−0.075	0.0016−0.0044

## Data Availability

Original data can be obtained by requesting the corresponding author (mtherrer@um.es).

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
