# Peer review of "Age and Sex Determine Electrocardiogram Parameters in the Octodon degus"

_biology, 2023, doi:10.3390/biology12050747_

Round 1
Reviewer 1 Report
Dear Authors,
First of all, I would like to congratulate you for the excellent scientific work carried out, as this work will certainly be very useful for researchers who, like me, study physiopharmacology of the cardiovascular system, however, I would like to ask you some questions that I believe are necessary:
1. In the literature, the association of ketamine with xylazine is frequent, however, you associated ketamine with medetomidine, why?
2. Have you or another research group already carried out experiments using, in the Octodon degus model, the association of ketamine with xylazine?
3. If yes, what were the results?
4. Were they similar to the results obtained with the combination of ketamine and medetomidine?
5. Have you already performed the experiments on this one using routinely used anesthetic agents such as inhalational anesthetics?
Kind regards,
Author Response
Dear Reviewer 1,
Thank you very much for the time and attention you dedicated to revise our manuscript. Please find in the attached Word document the answers to your comments.
Kind regards.

Reviewer 2 Report
Cardiovascular diseases represent the leading cause of morbidity and mortality, and age is an important risk factor. This study characterizes age-dependent changes in ECGs in the O. degus, a rodent model of aging. This study found that QRS duration significantly increases while heart rate decreases with age in O. degus model. In addition, there are significant sex differences in P, PR, QT, and the electrical axis of ECGs. Young males of o. degus exhibited higher incidences of arrhythmia. These age- and sex-dependent characteristics in ECGs indicate this animal model could be useful for studying age and sex dependent electrical remodeling.
Major comments:
Age-dependent electromechanical remodeling of the heart is an important topic and proper animal models of aging can be beneficial. O. degus is an interesting model that shows age-dependent deterioration in multi-organs such as brain and kidney. Thorough characterization of electrophysiology in this model is significant and findings of sex- and age-dependent changes in QT or other ECG parameters are interesting. However, the sample sizes vary quite a lot from n=4 to n=20, which concerns its statistical power, additional data points to the group with a small sample size are recommended. Although the discussion very nicely compares this rodent model to humans in aging and sex, it lacks comparison with other animal models such as mice, rats, rabbits, … etc. and missed the opportunity to highlight why O. degus is a unique model to study aging.
specific comments:
· Introduction: The introduction is quite brief compared to other sections, lacking rationale and justification to use O. degus. For example, line 77-89 says ‘… some of its similar age-related …’ but doesn’t say what ‘some’ here means. I recommend elaborating what age-related characteristics are known in this animal and why cardiovascular system can be also affected in this model.
· Group size: This is a major issue interpreting statistical power. Authors should choose the appropriate sample size. Only n=4 female juveniles were used in this study.
· QT vs. QTc: were QT measurements done without heart rate correction? If QT is heart rate corrected, QTc should be used instead.
· Figure 1, ECG example: Please replace the human ECG trace example with rodent ECG example to illustrate the existence of Q wave and how to measure QT interval in this rodent model.
· Table 4 from lines 225 – 230: I think ‘Table 4’ here means ‘Figure 4’ instead?
· Correlation analysis in Fig 2 E-H: I couldn’t fully understand the purpose and comparison. Were there any differences in these correlations among different sex or age groups?
· Fig 4 and line 224-231: I would move or copy the incidence numbers from the main text to the figure legend so that readers can easily correlate age/sex-differences in particular types of arrhythmia events.
· Typo in line 389: correct ‘EGC’
· Line 420, ‘some ECG values …’: please add what ECG values are changed.
· Line 425 – 430: If the sex-differences in arrhythmia is mainly caused by anesthesia and limited to ‘small rodents’, is this translatable to humans as well or not relevant to humans?
· Was there any age-dependent atrial arrhythmia in this animal model?
Author Response
Dear Reviewer 2,
Thank you very much for the time you dedicated to revise our manuscript. Please, find in the attached file the responses to your comments.
Kind regards.

Round 2
Reviewer 2 Report
The manuscript has been significantly improved and the authors addressed all of my concerns. The limitations addressed unequal or small number of samples and the introduction/discussion highlighted potential advantages of the proposed animal model over other models in studying aging. Just have one minor comment.
Line 474 'Atrio-ventricular conduction': I don't think increased P wave here indicates slow atrio-ventricular conduction. The following section addresses AV conduction issues through PR interval instead.
Author Response
Dear Reviewer 2:
Thank you very much again for your comments and for the time you dedicated to improve our manuscript. Please find here below the answers to your comment:
RESPONSES TO REVIEWER 2 COMMENTS ROUND 2
(MANUSCRIPT biology-2319758)
The manuscript has been significantly improved and the authors addressed all of my concerns. The limitations addressed unequal or small number of samples and the introduction/discussion highlighted potential advantages of the proposed animal model over other models in studying aging. Just have one minor comment.
Line 474 'Atrio-ventricular conduction': I don't think increased P wave here indicates slow atrio-ventricular conduction. The following section addresses AV conduction issues through PR interval instead.
Yes, you are right, PR interval is the parameter better reflecting the AV conduction. Please, see the revised version of the manuscript (lines 509-514). Thank you very much for your appreciation.
Kind regards.